# Mapping the Physical Language of Children Diagnosed with Autism: A Preliminary Study

**DOI:** 10.3390/children10071091

**Published:** 2023-06-21

**Authors:** Dita Federman, Adi Blustein, Tal-Chen Rabinowitch

**Affiliations:** The School of Creative Arts Therapies, University of Haifa, Haifa 3498838, Israel; ditafederman@gmail.com (D.F.); adiblustein@gmail.com (A.B.)

**Keywords:** autism spectrum disorder, siblings, children, physical language, self-touch

## Abstract

Children diagnosed with autism spectrum disorder have a unique motor profile, characterized by, for example, unusual posture or compulsive use of the body. However, not much is known about specific characteristics of their physical language, such as their movement direction, their self-touch pattern, etc., and even less is known about these characteristics with regard to their typically developing siblings. In this first of its kind study, we attempted to map the physical language of children diagnosed with autism spectrum disorder and to compare it to their typically developing siblings. To this end, we recruited 12 pairs of siblings, comprising one sibling with a diagnosis of autism and one sibling who is typically developing. The siblings were asked to play for 10 min and were videotaped throughout the interaction. We evaluated the siblings’ physical language using Laban’s movement analysis. We found significant and substantial differences between the physical language of the children diagnosed with autism and their typically developing siblings. The results are discussed in terms of the implications of the differences in physical language between the two populations and how movement analysis could be important for interventions in order to improve the communication and social abilities of ASD children.

## 1. Introduction

Autism spectrum disorder (ASD) is a neurodevelopmental disorder that is characterized by a triad of qualitative impairments in social interaction, communication, and in restricted, repetitive, and stereotyped behaviors [1]. Some studies have also shown that individuals with ASD do not demonstrate as much social motor coordination as their typically developing peers. This lack of coordination in both intentional or spontaneous interactions may limit the individual’s ability to socially and emotionally connect with others [2,3,4]. ASD can also be defined by restricted and repetitive behaviors and rituals, such as an unusual posture or compulsive use of the body, as portrayed in movements, such as incessantly turning around, as well as demonstrating repetitive speech and movements [1].

### 1.1. Behavioral Characteristics among Siblings of Children with ASD

Siblings of children with ASD have been studied primarily as a risk group, and until recently, little was known about sibling relationships in these families [5]. Given the prevalence of ASD, it can be estimated that in the last two decades, there has been a 300% increase in the number of children growing up alongside an ASD-diagnosed sibling [6]. Despite recent growing interest in sibling relationships in families of children with ASD, there is still much more ground for future studies to explore.

The literature on siblings of children with ASD provides inconsistent results, with some studies reporting them as more likely to have adverse outcomes than comparison groups and others reporting no significant differences.

Shivers et al. [7] showed that siblings of children with ASD have significantly more negative outcomes, such as negative beliefs (e.g., about themselves and about disabilities in general), more behavioral problems, worse psychological functioning, worse sibling relationships, and poorer social functioning than comparison groups, but there is no significant differences in adjustment, attention, hyperactivity, coping, or family functioning. In addition, sub-areas of functioning, in which siblings of children with ASD also performed worse, included symptoms of anxiety and depression [7]. Moreover, siblings of children with ASD had significantly lower functioning than siblings of individuals with intellectual and developmental disabilities as well as siblings without disabilities [7]. Therefore, studies of social functioning in siblings of children with ASD tend to operate under one of two hypotheses: (1) siblings of children with ASD will exhibit impaired social functioning [8,9], or (2) siblings of children with ASD are more likely to exhibit prosocial behavior and empathy, as their experiences at home motivate them to be more supportive and considerate of differences in others [10]. Taken together, although studies have examined several aspects of social functioning among siblings of children with ASD, it is not yet known whether they exhibit enhanced or impaired social skills.

### 1.2. Physical Characteristic and Motor Skills among Children Diagnosed with ASD

The mirror neuron system (MNS) is a brain network activated when we move our body parts and observe other agents’ actions [11]. Rogers and Penington [12] argued that an early impairment in the ability to imitate joins later difficulties in perceiving emotions, interpersonal relationships, and theory of mind, which together form the core of the difficulty in autism spectrum disorders. About 90% of the children who are diagnosed with ASD show impaired motor performance, which also includes motor imitation (e.g., Refs. [13,14]). The act of imitation requires synchronization between two partners, both in quality and in the operations’ time. The ability to imitate another’s action relies, among other things, on joint involvement and reciprocity [15,16]. Bontinck et al. [17] suggested that difficulties in coordination and synchronization among children with autism affect their ability to perform imitative actions. Synchronicity was found to be significant for the development of complex social behaviors, such as empathy and cooperation [15]. While imitation requires synchronization between two interacting partners, interpersonal synchrony does not necessarily require imitation. In addition, in a systematic review among individuals with ASD an association between social and motor skills was reported [18]. Difficulties in motor skills are found among siblings of children with ASD from the first years of life [19,20,21,22]. According to MacDonald et al. [22], fine and gross motor skills significantly predict calibrated autism severity. Children with weaker motor skills have more significant social communicative deficits [22]. Therefore, therapy that involves physical work and body movement may be especially relevant to this population.

### 1.3. Laban Movement Analysis

One of the most renowned measures to describe the function and expression of movement is Laban Movement Analysis (LMA), which was initially developed by Rudolf Laban [23], and has since been recognized and applied in numerous disciplines, including psychotherapy [24,25,26], sports and robotics [27]. LMA serves as one of the primary assessment tools in DMT [23,28,29,30,31,32].

LMA is an empirical, observational, and analytical system based on knowledge acquired through somatic and embodied practice. In LMA, movement is observed as a pattern of change in four components: body, effort, space, and shape (BESS). Briefly, the body category describes the structural and physical characteristics of the human body while moving. The effort category is also defined as dynamics, and it includes four factors (flow, weight, time, and space). Effort is used to understand the more delicate characteristics of movement concerning inner intention. The space category involves motions connecting with the environment and spatial patterns and pathways [33]. Through the shape category, the way the body figure changes during movement is further experienced and analyzed. Shaping occurs between bodies and is affected by the context, the quality of movement, and the participant’s beliefs and expectations [34]. Hence, LMA is a well-known methodology for categorizing and interpreting human movement. It has been used in the development of a movement curriculum and stressed embodiments of children and youth with ASD. LMA can help to compare movement patterns between the two groups in this study (i.e., ASD and TD), pointing to the differences in the use of body parts, postures, the direction of movement, the use of weight and self-touch.

### 1.4. The Current Study

The current study aims to map the children diagnosed with ASD and their TD siblings’ physical language.

Most studies on siblings of children with ASD focus on development, emotion, social, and cognitive implications and are conducted using qualitative methods: self-reports and parent and teacher reports. No studies have been conducted for siblings of children with ASD. This study is thus a preliminary attempt to map the physical language of children with ASD and compare it to their TD siblings.

## 2. Materials and Methods

### 2.1. Design

This work has a quantitative between-subjects research design. In order to evaluate children with ASD and their TD siblings’ physical language, children were observed at their home during two play episodes where both siblings were asked to play together (see below). We used video analysis to analyze the children’s physical language.

### 2.2. Research Instruments

Background data questionnaires were used to provide information on the age of the children, age of diagnosis, age of sibling participating in the study, and the characteristics of the parents (age, education, occupation, and socio-economic status of the family).

### 2.3. Observations

An experimenter video-recorded the siblings playing at their home for ten minutes. The time was divided equally between two parts: the first five minutes were a pre-planned specific activity—rolling a ball between the two siblings—and the last five minutes were free choice—the siblings could choose any activity they wanted to do together. The experimenter explained the siblings about the activities but did not interfere while they were playing together.

### 2.4. Movement Assessment Using Video

The body movement assessment tool in this study is based on concept of the Laban Movement Analysis Scale [35] and contains four factors: body attitude, shape, effort, and synchronization (see Table 1). In addition, we have also analyzed the children’s self-touch behavior.

Body attitude comprises the characteristic movement qualities that can be detected without formal movement notation [36]. It includes the use of body parts and readiness for certain patterns to emerge [28]. It refers to the characteristic body stance that is used and includes what is maintained or returned to. This is the type of readiness to act that is expressed in the trunk [28]. Body attitude and posture are often used interchangeably; posture generally incorporates expressive content [28]. Here, we identified whether the use of the children’s hand was either tight, which refers to holding the hand strongly closed as if it is difficult to release, or loose, which refers to holding the hand as if letting go, as well as whether the children’s general posture was either open, which refers to distancing body parts from each other or from the center of body outwards [37], or closed, where body parts are close to each other and/or to the center of body (Table 1).

Shape describes the changing forms that the body makes in space [38,39]. Kestenberg et al. [36] adds to Laban’s definition the term shape flow, which is the succession of changes in the shape of the body. Here, we identified the children’s movement direction, whether the movement was outward, also known as unfolding out or spreading, which refers to when the movement is directed outwards, away from center of the body or inwards, also known as folding in, which refers to when the movement is directed inwards, towards the center of the body (Table 1).

Efforts are movement qualities indicating mastery of the body and the environment [35]. Here, we identified the weight effort, which is the degree of strength used in action [40]. It is the attitude to the movement impact or intention, strong or light impact [41]. It is the way a person uses the weight of his/her body to cope with the environment. Weight is described as either light, which refers to the quality of movement that gives a gentle, light and weightless impression with very little grounding or strong, which involves putting an increasing pressure into movement [42], such as when stamping, clumping, or banging one’s feet against the floor [41]. Strong weight enables to perform activities that demand power. Laban connects strong weight with will-power and sticking to the goal (Table 1).

Synchronization in this sense refers to the integrated behavior of the individual [43], whether it is in body position, effort and gesture elements within oneself. Here, we identified the intra-synchronization between the children’s right and left hands (Table 1).

A time frame of ten minutes per observation is recommended as it permits adequate repetitions to enable the classification of habitual behaviors [44,45] and leaves the observer with an impression of a unique sequence of actions [46]. From the ten minutes that were recorded, the middle five minutes were selected for coding and analysis [47]. Two external observes coded each child participant on body movement features in (a) global analysis, if the child portrayed that specific behavior (i.e., score of 1) or has not (i.e., score of 0), and (b) frame-by-frame analysis, which is only for one dimension (i.e., self-touch).

Table 1 portrays the summary of the movement dimensions we identified and coded using global analysis.

Altogether, we evaluated 7 dimensions: 6 dimensions have been coded with global analysis, as shown in Table 1. In addition, 1 dimension, self-touch, which refers to the movement of touching the body with the palm of the hand was coded in a frame-by-frame analysis, due to the specific nature of this movement behavior. We identified 4 specific areas of self-touch: facial, which refers to the outer outline of the face, and making slight movement around the face using the hand; belly, which refers to touching the belly center with the palm and fingers; hand, which refers to hand-to-hand grip; and thigh, where one or both hands touch the thighs. We specifically chose to code every day, natural movements that we think might yield differences between children with ASD and TD. We also wanted to select movements that are somewhat easier to code so as to make this type of movement classification framework relevant and accessible for clients and practitioners.

### 2.5. Procedure

Participants’ parents were contacted and given a written explanation of the project aims and assessment procedure and were asked to sign a consent form. Subsequently, they filled out two questionnaires: (1) a background information questionnaire and (2) a Sibling Inventory of Behavior questionnaire (not reported in the current report). The observation time was coordinated with the parents during the hours when the children were used to playing and were not expected to be tired or hungry. When arriving at the house, the experimenter introduced herself to the children and explained that she is doing research on siblings and how they play together and asked the children to provide their oral assent prior to the beginning of the study.

Following oral assent, the experimenter said she wanted to see the children play, like they would have played if she had not videotaped them and explained that they could stop participating at any time if they so wished. The children were asked to choose a place where they regularly play. The specific instruction read: “You will play together with a ball for 5 min. Start on the ground; you can roll the ball to each other in every way that you want. After that, you will be free to choose any game that you like to play together, for 5 min”. The researcher positioned herself in a place that allowed her to observe the children without being seen, so that natural interaction between the two siblings could occur. The interaction was videotaped with parental permission.

### 2.6. Data Coding

Of the 10 min measured, the middle 5 min were selected for coding [47,48]. In studies on dyadic interactions, different coding methods are used, including global coding, in which a score or rating is given to a particular dimension in the interaction, and coding of the occurrence of the interaction moment by moment (frame-by-frame analysis). Global coding may be more suitable for examining questions and hypotheses regarding outcome measures and general questions, while micro-analytical coding is particularly suitable for the examining processes and questions regarding the amount or proportion of actions and situations, for example [17,49].

In the current study, coding was conducted both at a global level and at a micro-analytical level. Two dance movement therapists evaluated each video excerpt separately for each of the siblings. For the global coding, 6 features were coded for either “present” (=1) or “absent” (=0; see also Table 1), and for the frame-by-frame analysis, one feature was coded, as specified below.

### 2.7. Global Coding

Hand use—hand loosenessGeneral posture—openGeneral posture—closedMovement direction—inwardsWeight—lightSynchronization between right hand and left hand

### 2.8. Micro Analytic Coding

The micro-analytic coding included a frame-by-frame coding of self-touch for each sibling, and was formulated in three stages:Self-touch amount—the number of self-touch frames.Self-touch proportion—the number of self-touch frames divided by the total number of frames (steps 1 and 2 formulate the final *proportion* measure).Self-touch place (i.e., either belly, face, hand, or thigh).

### 2.9. Reliability between Judges

Cohen’s Kappa was run to determine if there was an agreement between the two raters regarding the 10 global features. Table 2 presents agreement measures of the 6 features between the two raters. Agreement ranged from 92% to 100%.

For the frame-by-frame self-touch analysis, agreement ranged between 87.5% and 100%. Cohen’s kappa was 0.97.

## 3. Results

### 3.1. Participants

Twenty-four children between the ages of 7 and 14 (M_age_ = 11.29, SD_age_ = 1.79) participated in the study; there were 12 pairs of siblings, each consisting of an ASD-diagnosed and a TD sibling. There population comprised 2 female dyads, 5 mixed-gender dyads, and 5 male dyads. In 10 out of the 12 dyads, the youngest sibling was the one diagnosed with ASD, whereas the other 2 dyads were twins. The mean age difference between the siblings was 2.58 years. The mothers’ mean age was 43.5 years (SD = 3.7); the fathers’ mean age was 46.8 years (SD = 4.2). The number of residents at home was either four (67%) or five (33%). The average income per month was either between ILS 10,000 and ILS 15,000 (33%) or above ILS 26,000 (67%); all families belonged to middle-high class according to the Israeli Knesset analysis of household income and expenses [50]. While 83% of the mothers have a Bacheller’s degree, 17% have a Master’s degree. Of the fathers, 92% have a Bacheller’s degree and 8% have a Master’s degree.

The participants’ sample consisted of two groups: (a) children diagnosed with ASD (2 girls and 10 boys); this gender ratio is compatible with the reported distribution of ASD in Israel [51,52], and (b) children who are typically developing (5 boys and 7 girls), without a diagnosis of ASD or another developmental, neurological, or chronic health disability, according to parental report.

### 3.2. Physical Language

Table 3 portrays the summary of the differences between the two children (ASD and TD) in the specific physical language characteristics that we identified and coded (see Table 1). A higher percentage score means more children had executed that specific physical language characteristic, whereas a lower percentage score denotes a smaller number of children executing that specific physical language characteristic. We evaluated the differences between the ASD and TD siblings using McNemar’s test (as the two observations are paired; Table 3).

High percentage indicates higher occurrence rate, whereas a lower percentage indicates a lower occurrence rate of that specific physical language characteristic.

To summarize, the analysis shows a distinct and statistically significant difference between the specific features of the physical language of children diagnosed with ASD and their TD siblings.

### 3.3. Self-Touch

Table 4 presents the frame-by-frame self-touch analysis. Frame-by-frame differences of proportions of self-touch body parts by groups (ASD vs. TD) was performed using SAS’s version 9.4 for windows procedure PROC GLIMMIX. We referred the two measurements measured per family (ASD and TD) as repeated measures. A one-way repeated-measures analysis of variance (ANOVA) with one within-family variable (group: ASD vs. TD) and age of child as a covariate, to control for age differences, was employed. The GLIMMIX procedure enables fitting various distributions of the response variable and considers the two-level hierarchical structure of the data: two siblings per family.

Overall, children diagnosed with ASD touch themselves significantly more than typically developing children. For specific body parts, statistically significant differences are established only for the hand and thigh touch.

## 4. Discussion

Overall, we found significant differences between the physical language of children diagnosed with ASD and their typically developing siblings in a number of characteristics, such as their hand use, posture, movement direction, weight distribution and the level of synchronization between their hands. Moreover, it was found that children who are diagnosed with ASD touch themselves significantly more than their typically developing siblings.

### 4.1. Physical Language

The physical language of children diagnosed with ASD was found to be different from that of their TD siblings. Understanding and controlling motor actions is an essential aspect of daily life. The body posture changes depending on the self and/or the environment [33]; a closed body posture, directed inward, may indicate difficulty in creating contact with the external world, while an open physical position, directed outward, may indicate a willingness to create contact with the world. These patterns correspond with the findings from the present study: children who are diagnosed with ASD are characterized by inward movements and a closed body posture, while their TD siblings are characterized by outward movements and an open body posture, with substantial weight and hand holding. These findings agree with their current developmental stage and their yearning for contact with the world and regard for the environment [54,55,56]. In contrast, the body language of the children diagnosed with ASD is directed inwards: inward movement, lightweight, closed body posture, and hand looseness. These correspond with deficiencies and difficulties in establishing contact and communication with the environment [33] and might coincide with withdrawal into their inner world [1,57]. In addition, the lack of synchronization between the two hands for the diagnosed children signifies a lack of integration within oneself [43], which may be essential in order to sustain integration and interpersonal synchrony with another person. As much as these findings agree with the developmental level of the two groups of children, it is also possible that some of these differences in physical language may be attributed to the difference in age, as the diagnosed children’s cohort is younger than their TD siblings. Future research will need to determine if at all and how much age plays a role in these specific physical language behaviors.

### 4.2. Self-Touch

Self-touch, like mannerisms and posture shifts, is a type of self-regulatory movement [58,59,60,61]. Self-touch movements are generally performed with one or both hands and involve rubbing, scratching, or holding any part of the body’s surface or pieces of clothing [62,63,64,65]. Recent studies support the claim that self-touch is an essential part of self-regulation skills that emerge early with significant effects on socio-emotional and cognitive developmental trajectories of infants [66,67], which is deficient in individuals diagnosed with ASD [3,68]. Our research focuses on four self-touch body parts: (1) facial self-touch (2) hands self-touch (3) belly self-touch and (4) thighs self-touch. Self-touch serves the regulation of both hyper- and hypo-arousals. Although, it remains unclear if different forms of self-touch occur in different contexts and if the regulatory mechanisms are acquired or innate [69,70]. In the current study, a significant effect was found in the frequency of self-touch between the two groups. Children diagnosed with ASD showed more self-touch behavior than their typically developing siblings. This effect has been observed in self-touch overall, as well as in the hand and thigh self-touch. Some of the most identified symptoms of ASD are regulation difficulties [68]. The need for regulation, which is more present in ASD compared to TD children, and an early developmental stage, are consistent with the differences in the frequency of self-touch between the two groups. While touching one’s own hands is relatively fairly common, self-touch in the thighs is not generally common and was observed with high frequency among the ASD group. The thighs are the part of the body located between the hip and the knee, and this location might be connected with the genital area due to their close proximity. Individuals diagnosed with ASD have similar sexual desires and needs when compared to their neuro-typical peers [71,72]. Like other aspects of development, sexual development occurs through stages across the individual’s life span and continues to evolve from birth to death [73]. Childhood sexual development, from birth to seven years, is marked by curiosity and social rules, and contains behaviors, like sexual play, self-touch, and gender identification [73]. Children diagnosed with ASD may be more likely to demonstrate sexual behaviors as they struggle to navigate the physical, emotional, and social challenges inherent in exploring their sexuality. There are multiple parent report-based studies where parents and caregivers indicate high rates of problematic sexual behaviors of their children who are diagnosed with ASD [72,74,75,76,77], such as inappropriate self-touch [71]. It is important to note that the thighs self-touch in this specific study was not experienced as inappropriate by the observers but might appear a bit unusual and repetitive when considered in a sexual context. Also, the observation was videotaped by the researcher, who is not a member of the family, and only for ten minutes, which might not allow enough time for the children to feal at ease, and perhaps that is why they exhibit more self-touch. Thus, there is a need for further research to allow a longer, more reliable and comprehensive observation in this direction.

To the best of our knowledge, this is the first study to examine the physical language of children on the autistic spectrum compared to the physical language of their typically developing siblings, albeit work that had been conducted in other modes or channels, such as speech, hearing, gestures and more [78]. Previous studies that used Laban Movement Analysis (LMA) with ASD-diagnosed individuals, focused on movement and dance as an intervention technique [79,80,81], while the current study focused on the mapping of the physical language of ASD-diagnosed children. The use of LMA in this study provided a systematic way to observe, recognize and describe patterns of change. It interprets, and understands movement, and uses approaches, based on anatomy, kinesiology and psychology. Movement is observed as a pattern of change that occurs in terms of four components, defined as body, effort, space and shape. While the Laban Movement Analysis offers valuable insights into human movement, it also has some limitations. For example, it does not analyze all forms of human movements, as body movements vary according to different cultures and the use of movements varies from culture to culture, which could create a cultural bias. It should be taken into account that LMA was developed within a Western culture and may not meet the nuances of other cultures [82]. Body structures that have not been addressed by LMA are body structures of people with disabilities, footsteps of dance tradition and eye movements in dance tradition [82]. In addition, there may be discrepancies in the analysis due to the observer’s interpretation and subjective judgment, as such subjectivity may be a major limitation. It is therefore advised to use two expert judgments (as conducted in this study), which could be compared for reliability.

This work was an initial attempt to map the physical language of children diagnosed with ASD and their TD siblings to find movement patterns that can reveal how children carry themselves in the world in general. The research setting was created to enable the collection of as much information as possible on a topic for which knowledge is relatively slim. The study’s findings show that there are repeated and shared patterns of movement among each group. The results of the current study highlight that much attention must be given to physical language and movement, which according to Laban and Lawrence [33], both produces and expresses our mental and emotional being. Consequently, we hope that once more research and findings start to emerge in this area, body movement mapping could be included in the diagnosis, treatment, and development of intervention methods that will contribute to a confident and beneficial interaction between siblings and improve the communication and social abilities of children diagnosed with autism spectrum disorder.

## Figures and Tables

**Table 1 children-10-01091-t001:** Movement dimensions evaluated by external observes for global analysis.

Body Attitude			Shape	Effort	Interpersonal Synchrony
Hand use	General posture		Movement direction	Weight	Synchrony between hands
hand looseness	open	closed	movement inwards	light	sync between right and left hand

**Table 2 children-10-01091-t002:** Agreement of ten features between two raters.

Feature	Kappa	Z	*p*-Value	Agreement
Hand looseness	1.00	4.90	<0.0001	100.00
General posture—open	1.00	4.90	<0.0001	100.00
General posture—closed	1.00	4.90	<0.0001	100.00
Movement—inwards	1.00	4.90	<0.0001	100.00
Weight—light	0.92	4.51	<0.0001	95.83
Synchronization between right and left hands	1.00	4.90	<0.0001	100.00

**Table 3 children-10-01091-t003:** The occurrence of specific physical language characteristics in ASD and TD siblings.

Type of Behavior	TD	ASD	X^2^(DF)	*p*-Value
Hand looseness	0%	100%	24(1)	0.001
General posture—open	83.3%	0%	17(1)	0.004
General posture—closed	16.7%	100%	17(1)	0.004
Movement—inwards	0%	100%	24(1)	0.001
Weight—light	0%	100%	24(1)	0.001
Synchronization between right hand and left hand	100%	0%	24(1)	0.001

**Table 4 children-10-01091-t004:** Comparison of self-touch proportions between ASD and TD.

Self-Touch Body Part	ASD, *n* = 3600 Frames	TD, *n* = 3600 Frames	F(1,22)	*p*-Value	Partial η^2^
Belly	103 (0.90%)	17 (0.34%)	3.45	0.09	0.13
Face	123 (1.96%)	57 (1.60%)	0.44	0.52	0.02
Hand	44 (0.90%)	13 (0.01%)	10.99	0.008	0.32
Thigh	535 (16.76%)	96 (2.10%)	103.23	<0.0001	0.82
Self-touch overall	846 (22.70%)	185 (5.00%)	116.4	<0.0001	0.84

Note: partial η^2^, effect size, according to Cohen, J. [53] considered 0.01 effect as small, 0.06 considered medium and 0.14 considered large; self-touch proportion was adjusted to child’s age.

## Data Availability

Data are available from the corresponding author upon reasonable request. The analyses presented here were not preregistered.

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
