# Peer review of "Mapping the Physical Language of Children Diagnosed with Autism: A Preliminary Study"

_children, 2023, doi:10.3390/children10071091_

Round 1

Reviewer 1 Report

This study aims to analyze motor patterns in children with autism spectrum disorder and in their siblings, in order to compare them and to detect any differences. However, the paper is difficult to be read since it presents a discrepancy between a verbous first part, whose information is often not useful to the study itself and to undestand the conclusions, and the core of the article.

In particular, the introduction appears excessively detailed and it provides numerous hints that are not relevant for this study, and not suitably interrelated. It is not clear why the Laban movement analysis has been adopted in place of other standardized scales. Furthermore, the paragraph about Dance/Movement Therapy appears not necessary, since its concepts have not been developed in the study.

The Discussion is overly vague and superficial, and does not explain how to use and interpret the obtained data in terms of screening, tools for diagnosis or therapy.

As a side note, Table 1 is not easy to be understood, also due to font inaccuracies (i.e. wrong use of bold).

Author Response

This study aims to analyze motor patterns in children with autism spectrum disorder and in their siblings, in order to compare them and to detect any differences. However, the paper is dif<icult to be read since it presents a discrepancy between a verbous <irst part, whose information is often not useful to the study itself and to undestand the conclusions, and the core of the article.

In particular, the introduction appears excessively detailed and it provides numerous hints that are not relevant for this study, and not suitably interrelated.

Reply: We have now done extensive re-editing of the Introduction. Please see revised manuscript. In addition, we completely deleted section 1.4 about Dance-Movement Therapy (as suggested in the reviews).

It is not clear why the Laban movement analysis has been adopted in place of other standardized scales.

Reply: We have now re-edited (the new) section 1.4 about Laban Movement analysis and speci<ically explained in the text why we chose LMA over standardized scales (lines 185-190).

Furthermore, the paragraph about Dance/Movement Therapy appears not necessary, since its concepts have not been developed in the study.

Reply: Yes, indeed. We have now completely deleted this section from the text.

The Discussion is overly vague and super<icial, and does not explain how to use and interpret the obtained data in terms of screening, tools for diagnosis or therapy.

Reply: We will not be concentrating on Dance-Movement Therapy in this paper therefore will not be getting into practice issues, etc. Indeed, this is a basic research paper, describing the mapping of body language between ASD children and their siblings.

As a side note, Table 1 is not easy to be understood, also due to font inaccuracies (i.e. wrong use of bold).

Reply: we have now re-edited Table 1. We hope it is clearer.

Reviewer 2 Report

This study applied Laban Movement Analysis (LMA) to code videos that reflect the physical movements of children diagnosed with autism spectrum disorder (ASD) and their typically developing siblings. The researchers found significant differences between the two groups and argued for the need to consider interventions such as Dance-Movement Therapy to improve the behavioral patterns in children with autism. 

The manuscript showed a good coverage of the related literature and presented an interesting perspective to study movement patterns in autism. There are several areas that need to be improved before I would recommend acceptance.

1. The Introduction section 1.4 on Dance Movement Therapy seems a little disconnected with the overall conceptual flow to motivate the research questions unless testing the efficacy of the therapy is the research objective. It is unclear whether the children included in the study was enrolled in a Dance Movement Therapy program or not. If yes, how long they have had the therapy. It also seemed tautological when only behavioral patterns were compared for those diagnosed with autism and those without. It would be surprising that these two groups would demonstrate similar behavioral patterns using the LMA analysis. 

2. Several aspects of video recorded behavior can be potential targets for analysis regarding movement direction (repetitive or stereotyped movements such as hand flapping, rocking or spinning), gesture usage, facial expressions and eye contact, motor coordination in fine or gross motor tasks, sensory responses, and overall body language in terms of level of engagement, social interaction and fluidity of movements. LMA provides a comprehensive system for observing, describing, and interpreting human movement, including aspects of body, effort, space, and shape. LMA can help in identifying and comparing movement patterns between the two groups, highlighting differences in movement qualities, postures, gestures, and self-touch patterns. It can also provide insights into motor coordination, spatial awareness, and overall body language. However, it's important to consider that LMA is primarily focused on movement analysis and may not fully capture all key aspects of behavior relevant to autism. It may need to be supplemented with other coding systems or observational methods specific to autism research to encompass a broader range of behaviors associated with ASD. Furthermore, when using LMA to analyze movement behavior in children with autism spectrum disorder (ASD) and their typically developing siblings, it is crucial to consider the specific challenges and characteristics associated with ASD. Maybe the discussion needs to go deeper into what the LMA is good at and what its limitations are. Please note that LMA and DMT had been used in previous autism studies, which should be cited. For instance,

https://www.pdcnet.org/phenomenology2010/content/phenomenology2010_2010_0105_0132

http://is.mpg.de/uploads_file/attachment/attachment/449/Bevill16-ROMANWS-Behavioral.pdf

https://hdl.handle.net/2292/54998

https://www.mdpi.com/2076-328X/7/1/14

3. A major concern is the statistical analysis of the data. 

3a. McNemar's test is a statistical test used to analyze paired nominal data, where the same subjects are observed under two different conditions or treatments. It is typically used when the outcome variable is dichotomous (e.g., presence/absence of a behavior). In the context of comparing multiple behaviors between an autism group and a control group, McNemar's test may not be the most appropriate statistical test because McNemar's test is specifically designed for analyzing the changes in a single variable between paired observations, rather than comparing multiple variables simultaneously. It is advisable to consult with a statistician or data analyst who can assess your specific research design, data, and objectives to determine the most appropriate statistical tests and consider any necessary corrections for multiple comparisons.

3b. Behaviors such as hand holding vs. hand looseness, posture open vs. posture closed, movement inwards vs. outwards, weight light vs. weight strong, synchronization between two hands vs. asynchronization are expected to be highly negatively correlated in the coding data. The tables reporting the statistical results seemed redundant. Instead of analyzing each behavior separately, you could consider creating composite variables or indices that capture the overall pattern or style of movement. This can help reduce the issue of multiple comparisons and provide a more comprehensive representation of the movement profile. For example, you could create a "synchronization-asynchronization index" to capture the coordination between the two hands. By creating composite variables, you can examine the overall relationship between these movement patterns and the groups (autism vs. control) without inflating the number of statistical tests conducted. This can provide a clearer picture of the movement characteristics associated with each group and potentially identify meaningful differences. 

3c. Self-touch behaviors for different body parts may be correlated, assuming that individuals tend to engage in self-touch across multiple body parts in a similar manner. If so, it may not be necessary to analyze each body part separately. Instead, you can consider creating a composite variable or index that captures the overall level or frequency of self-touch behavior across all body parts. As you have both the number of touches and the duration of self-touch measures available, you need to analyze them separately because they represent different aspects of self-touch behavior. The number of touches represents the frequency or count of self-touch events, while the duration represents the length of time spent in self-touching behaviors. Additionally, it's worth considering that the number of touches and duration of self-touch may not be completely independent measures. They could be correlated, and accounting for this correlation in the analysis could enhance the accuracy of the results. This can be achieved through appropriate statistical modeling, such as using multivariate regression or mixed-effects models. In this experimental design where the scores are paired by family with one member being autistic and the other sibling not autistic, it appears the use of ANOVA is problematic. 

3d. In the analysis of inter-rater reliability, Cohen's Kappa is suitable for categorical or nominal ratings where the data is divided into discrete categories. But self-touch behavior in terms of duration included continuous variable. Maybe Intraclass Correlation Coefficient (ICC) should be considered as it is a versatile measure of reliability that can be used for continuous, ordinal, or interval level ratings. 

3e. Even though the sample is very small, it is worth splitting the pairs into two subgroups in which one subgroup had the older sibling diagnosed with autism and the other subgroup had the younger sibling with autism to see the age effects. Again, more complicated modelling may be necessary. 

4. If the authors think it is important to talk about how emotions are recognized in different channels. Please consider citing updated review literature involving children with autism. For example,

https://doi.org/10.1177/1362361321995725

https://doi.org/doi:10.1044/2022_JSLHR-21-00438

5. How do the results using LMA coding for physical language correlate with other measures of autism? What are the limitations and future directions? For instance, it seems unclear why the Dance Movement Therapy was mentioned in the introduction to motivate the research and yet hardly discussed in the end. 

Author Response

This study applied Laban Movement Analysis (LMA) to code videos that re<lect the physical movements of children diagnosed with autism spectrum disorder (ASD) and their typically developing siblings. The researchers found signi<icant differences between the two groups and argued for the need to consider interventions such as Dance-Movement Therapy to improve the behavioral patterns in children with autism.

The manuscript showed a good coverage of the related literature and presented an interesting perspective to study movement patterns in autism. There are several areas that need to be improved before I would recommend acceptance.

Q1. The Introduction section 1.4 on Dance Movement Therapy seems a little disconnected with the overall conceptual <low to motivate the research questions unless testing the ef<icacy of the therapy is the research objective. It is unclear whether the children included in the study was enrolled in a Dance Movement Therapy program or not. If yes, how long they have had the therapy. It also seemed tautological when only behavioral patterns were compared for those diagnosed with autism and those without. It would be surprising that these two groups would demonstrate similar behavioral patterns using the LMA analysis.

Reply: The aim of the study was to map children’s movements. The children were not enrolled in a Dance Movement Therapy program. We apologize for the confusion. We have now deleted the part about Dance Movement Therapy from the manuscript.

It is probably right that it would make sense that these two groups would demonstrate distinct movement patterns, and indeed, our goal was to map speci<ic every day, natural movement patterns and provide an interpretation within the context of the differences between the ASD children and their TD siblings (see lines 420-423).

Q2. Several aspects of video recorded behavior can be potential targets for analysis regarding movement direction (repetitive or stereotyped movements such as hand <lapping, rocking or spinning), gesture usage, facial expressions and eye contact, motor coordination in <ine or gross motor tasks, sensory responses, and overall body language in terms of level of engagement, social interaction and <luidity of movements. LMA provides a comprehensive system for observing, describing, and interpreting human movement, including aspects of body, effort, space, and shape. LMA can help in identifying and comparing movement patterns between the two groups, highlighting differences in movement qualities, postures, gestures, and self-touch patterns. It can also provide insights into motor coordination, spatial awareness, and overall body language. However, it's important to consider that LMA is primarily focused on movement analysis and may not fully capture all key aspects of behavior relevant to autism. It may need to be supplemented with other coding systems or observational methods speci<ic to autism research to encompass a broader range of behaviors associated with ASD. Furthermore, when using LMA to analyze movement behavior in children with autism spectrum disorder (ASD) and their typically developing siblings, it is crucial to consider the speci<ic challenges and characteristics associated with ASD. Maybe the discussion needs to go deeper into what the LMA is good at and what its limitations are. Please note that LMA and DMT had been used in previous autism studies, which should be cited. For instance,
https://www.pdcnet.org/phenomenology2010/conten/phenomenology2010_2010_0105_0132
http://is.mpg.de/uploads_<ile/attachment/attachment/449/Bevill16-ROMANWS-Behavioral.pdf
https://hdl.handle.net/2292/54998
https://www.mdpi.com/2076-328X/7/1/14

Reply: We have now added an elaborated section in the Discussion that delineates both other studies that used LMA with ASD diagnosed individuals, as well as possible limitations of using LMA in this current study (lines 649-668).

Q3. A major concern is the statistical analysis of the data.
3a. McNemar's test is a statistical test used to analyze paired nominal data, where the same subjects are observed under two different conditions or treatments. It is typically used when the outcome variable is dichotomous (e.g., presence/absence of a behavior). In the context of comparing multiple behaviors between an autism group and a control group, McNemar's test may not be the most appropriate statistical test because McNemar's test is speci<ically designed for analyzing the changes in a single variable between paired observations, rather than comparing multiple variables simultaneously. It is advisable to
consult with a statistician or data analyst who can assess your speci<ic research design, data, and objectives to determine the most appropriate statistical tests and consider any necessary corrections for multiple comparisons.

Reply: Thank you for pointing this out. We indeed consider the two observations (ASD and control) as a pair rather than two independent group observations since there are two siblings in the same family (ASD and control). Due to this, we used the McNemar’s test instead of a two-sample test. We added a clarifying note about this in lines 554-555.

3b. Behaviors such as hand holding vs. hand looseness, posture open vs. posture closed, movement inwards vs. outwards, weight light vs. weight strong, synchronization between two hands vs. asynchronization are expected to be highly negatively correlated in the coding data. The tables reporting the statistical results seemed redundant. Instead of analyzing each behavior separately, you could consider creating composite variables or indices that capture the overall pattern or style of movement. This can help reduce the issue of multiple comparisons and provide a more comprehensive representation of the movement pro<ile. For example, you could create a "synchronization-asynchronization index" to capture the coordination between the two hands. By creating composite variables, you can examine the overall relationship between these movement patterns and the groups (autism vs. control) without in<lating the number of statistical tests conducted. This can provide a clearer picture of the movement characteristics associated with each group and potentially identify meaningful differences.

Reply: Thank you for this suggestion. We have now revised our categories and reduced the number of variables to avoid being redundant. Speci<ically, instead of reporting both ends of the same behavior (e.g., movement inwards vs. outwards, etc.), we have chosen to only report one behavior instead of both behaviors. Please see revised Tables 1-3 and also in the text.

3c. Self-touch behaviors for different body parts may be correlated, assuming that individuals tend to engage in self-touch across multiple body parts in a similar manner. If so, it may not be necessary to analyze each body part separately. Instead, you can consider creating a composite variable or index that captures the overall level or frequency of self-touch behavior across all body parts. As you have both the number of touches and the duration of self-touch measures available, you need to analyze them separately because they represent different aspects of self-touch behavior. The number of touches represents
the frequency or count of self-touch events, while the duration represents the length of time spent in self-touching behaviors. Additionally, it's worth considering that the number of touches and duration of self-touch may not be completely independent measures. They could be correlated, and accounting for this correlation in the analysis could enhance the accuracy of the results. This can be achieved through appropriate statistical modeling, such as using multivariate regression or mixed-effects models. In this experimental design where the scores are paired by family with one member being autistic and the other sibling not autistic, it appears the use of ANOVA is problematic.

Reply: By itself, the number of touches has no meaning unless divided by the total number of frames. The result of this division was mistakenly called duration rather than proportion of self-touch. There are no two measurements available, only one – the proportion of self-touch –and this proportion is compared between the groups using SAS's version 9.4 for windows procedure PROC GLIMMIX, which is a procedure that employes mixed models to binomial variables like here (proportion). We did, as you suggested, refereed the two measurements of the two siblings (ASD and TD) per family as repeated measures. Please see our speci<ic change in the text in lines 451-452 and 476-479.

3d. In the analysis of inter-rater reliability, Cohen's Kappa is suitable for categorical or nominal ratings where the data is divided into discrete categories. But self-touch behavior in terms of duration included continuous variable. Maybe Intraclass Correlation Coef<icient (ICC) should be considered as it is a versatile measure of reliability that can be

Reply: In self touch the rating is touch/no touch and not a duration (please see comment 3c, this review). This is why we used the KAPPA test for inter rater reliability and not ICC. We hope that now it’s clearer.

3e. Even though the sample is very small, it is worth splitting the pairs into two subgroups in which one subgroup had the older sibling diagnosed with autism and the other subgroup had the younger sibling with autism to see the age effects. Again, more complicated modelling may be necessary.

Reply: Thank you for this suggestion. However, in this case, the diagnosed sibling was either always the younger sibling or a twin, so the variability in this case is extremely small if at all. Please also see lines 493-494 in the manuscript.

Q4. If the authors think it is important to talk about how emotions are recognized in different channels. Please consider citing updated review literature involving children with autism.
For example,
https://doi.org/10.1177/1362361321995725
https://doi.org/doi:10.1044/2022_JSLHR-21-00438

Reply: Thank you for this suggestion. As these studies are about the perception of children with ASD and not their production, we feel it is a bit outside the scope of the current paper, as it stands now.

Q5. How do the results using LMA coding for physical language correlate with other measures of autism? What are the limitations and future directions? For instance, it seems unclear why the Dance Movement Therapy was mentioned in the introduction to motivate the research and yet hardly discussed in the end.

Reply: We have now deleted the Dance-Movement Therapy section from the introduction. Thank you for pointing this out. We have also added a section that discusses the limitations of LMA (lines 649-668).

Round 2

Reviewer 1 Report

After revision process, the article is more fluid and the aim of the study, the results and the discussion are now more intelligible. I suggest to delete the paragraph 1.1. Challenges among ASD families since it is not useful to the discussion.

English language revision should be done.

Author Response

We would like to thank both reviewers for taking the time to provide more helpful comments on the manuscript.

Please see attached our point-by-point responses.

Reviewer 2 Report

1. The experimental design has some confounding issues.  Age is clearly an important factor in developmental research. If the autism child is younger than the sibling without autism, there is an age bias in favor of the older sibling, which would be different from the case of twins. The authors did not properly take into account the potential age effect in statistical analysis and interpretation. 

2. The manuscript needs to highlight the changes in the revision. 

3. LMA has been used in autism research before. There are also articles on how to use LMA for video analysis and how the statistical results are interpreted. Here I gave some examples. Please consider include those relevant to the current study in the Introduction and Discussion. 

https://www.mdpi.com/2076-328X/7/1/14/htm

https://www.ncbi.nlm.nih.gov/pmc/articles/PMC6455080/

https://www.frontiersin.org/articles/10.3389/fpsyg.2019.00572/full

Author Response

(The authors gave the same response as above.)
